# The impact of latent tuberculosis screening programmes for migrant populations in high income, low burden countries

Alice E. Taylor[1], Eisin McDonald[1], Hazel Henderson[1], Peter MacPherson[1,2]*

1 Public Health Scotland, Edinburgh, Scotland, 2 School of Health and Wellbeing, University of Glasgow, Glasgow, Scotland

* peter.macpherson@glasgow.ac.uk

## Abstract

### Background

Migrants from high to low tuberculosis (TB) incidence countries may benefit from screening for latent TB infection (LTBI), but the optimal approaches and effectiveness are not well described.

### Methods

Our primary objective was to synthesise evidence for cost-effectiveness, and barriers and facilitators to successful implementation, of LTBI screening programmes for migrants entering high income, low TB burden countries. Evidence was synthesised using rapid review methodologies.

### Results

41 studies (including 2 reviews) were included, covering the European region and national programmes. Main settings of LTBI screening were primary care, new arrivals clinics, and schools. The most frequently cited facilitator was structural cohesiveness (integration of health care services, collaboration with community partnerships, and co-ordination of care with social workers or accommodation staff). The most frequently cited barrier was lack of understanding and misconceptions of service users. Economic evaluations consistently demonstrated long term cost-savings for LTBI programmes. Screening migrants from countries of origin of the highest TB burden is more cost-effective but less likely to identify all TB and ultimately eliminate TB compared to screening at a lower TB burden threshold.

**Data availability statement:** All relevant data are within the paper and its Supporting Information files.

**Funding:** PM is funded by Wellcome (304666/Z/23/Z). For the purpose of open access, the author has applied a CC BY public copyright licence to any Author Accepted Manuscript version arising from this submission. The funders had no role in study design, data collection and analysis, decision to publish, or preparation of the manuscript.

**Competing interests:** The authors have declared that no competing interests exist.

## Conclusions

We found consistent evidence that LTBI screening programmes for migrants from high to low tuberculosis TB incidence countries can be effective and cost-saving in a variety of settings. A co-ordinated, integrated approach is a key programme facilitator.

## Introduction

Tuberculosis (TB) remains a major global public health problem, despite being preventable and treatable. The World Health Organization (WHO) set out it's End TB Strategy in 2014 to reduce TB incidence by 90%, TB deaths by 95%, and to eliminate catastrophic costs for TB-affected households by 2035 [1]. The TB pre-elimination phase is defined by the WHO as <10/1,000,000 new TB cases per year [2]. In 2023, 60 countries, predominantly in The Americas and European WHO regions, reported TB incidence of <10/100,000, and are thereby well placed to achieve TB elimination [3]. In 2023, there were 1.25 million deaths due to TB, likely returning to the leading cause of death due to an infectious agent worldwide, following the COVID-19 pandemic. There were 10.8 million people notified with TB in 2023, 12% of whom were children and young adolescents [3].

In 2014, 23% of people were estimated to have been exposed to *Mycobacterium tuberculosis* (MTB) and would have "MTB immunoreactivity" if tested with tuberculin skin tests or interferon gamma release assays [4]. Recent insights into the dynamic nature of transition between TB states emphasises that most TB disease is due to recent infection, with 2% of the global population recently infected with MTB and at greatest risk of disease, and 87% residing in the South-East Asia, Western-Pacific and Africa WHO regions [4–7]. Therefore, people from high TB incidence countries (and especially young children, and people with other risk factors) with immunological evidence of recent MTB infection (hereafter called LTBI), are likely to benefit from LTBI screening and provision of TB preventive therapy (TPT) [8].

International guidance from the WHO and the European Centre for Disease Prevention and Control (ECDC) recommend consideration of systemic screening for certain at-risk groups for LTBI, including people who have migrated from countries with a high TB burden [9,10]. These guidance have been variably implemented in countries with a low incidence of TB. For example, in Scotland, Wales and England, where numbers of people diagnosed with TB increased by 40.8%, 18.7% and 11.0% from 2022 to 2023 respectively, there is national clinical guidance in place recommending the screening of migrants for LTBI from countries with an incidence of >150/100,000 people [11–14]. Whilst an LTBI screening programme has been in place for new entrants to England and Wales since 2015/16, there is no such programme in place in Scotland, where the largest increase across the three nations has been seen [15]. In Scotland, 67.5% of diagnoses were in people born outside of the UK and the most commonly reported risk factor for a TB diagnosis was being a refugee or asylum seeker, with a shift over the last decade from alcohol misuse use as the most frequent risk factor [13].

Implementation of LTBI screening programmes targeted towards migrant populations from high TB incidence countries are an important component of a comprehensive public health approach to achieving TB elimination. LTBI screening programmes for migrant populations require multisectoral interventions, including systematic identification of new migrants from high TB burden countries who may be at risk of TB; screening for LTBI, and supporting TPT initiation and care; and addressing the social determinants of TB. Targeting of screening resources towards this group could also help to address the widening disparities in TB incidence.

The optimal programmatic approaches and cost-effectiveness of LTBI screening programmes have not been well described. Synthesis of existing evidence could support low TB incidence countries such as Scotland to implement effective programmes and accelerate towards elimination of TB. We therefore conducted a rapid literature review to understand how countries with a low TB burden, high income and high net migration are currently approaching screening for LTBI in migrant populations. We synthesised data on implementation approaches, barriers and facilitators to successful programmes, epidemiological impacts, and economic outcomes.

## Methods

Rapid review methodologies were used to synthesise available evidence for LTBI screening programmes, in order to facilitate the provision of timely insights to inform policy. Rapid reviews are a relatively novel methodology, used to synthesise evidence in a systematic, resource efficient manner [16]. A search strategy was developed and agreed upon by the research team prior to literature search. Articles were screened at title, abstract and full text stage by a single reviewer (AT), with a second review by PM where requested. A predefined data extraction table was designed and populated in Microsoft Excel, using the key areas of study characteristics, epidemiology, economic evaluations and barriers and facilitators. AT quality assessed individual studies using the appropriate CASP checklist according to the study design [17]. Medline, ProQuest Public Health, Scopus, bibliographic databases were searched by a public health librarian. A grey literature search was conducted, including the Social Policy and Practice database and Government websites of the countries as limited by the above conditions (full list of grey literature sources can be seen in S1 Appendix). References were imported into the reference manager SciWheel for de-duplication and processing.

Inclusion criteria for country of programme were (1) high income countries as defined by the World Bank Group for the 2023/24 fiscal year [18], (2) low TB burden countries (<10/100,000 population) as defined by the World Health Organization 2023 TB burden estimates [19], (3) High net in-migration in 2021 (>10,000 migrants/year) as per the World Bank Group [20] and (4) publication during or after 2000, with evaluation of quantitative and/or qualitative data, collected in part or whole, during or after 2000. Eligible studies evaluated a national or large co-ordinated screening programme, derived from or informing national screening programme guidance, policy or recommendations either in part or whole aimed at migrant populations. Exclusion criteria included smaller ad-hoc studies, studies that were not specific to migrant populations or latent TB, pre-migration screening studies and exclusively port-of-arrival studies. Pre-entry screening programmes were excluded. No limitations were placed on language. The search strategy can be seen in S1 Appendix.

From included studies, we extracted and summarised using narrative synthesis data from three key thematic areas. (1) Logistics, including: data collection period, location, settings, supporting institutions and organisations, community engagement. (2) Implementation of LTBI screening programme: screening target population/s, screening tools, clinical services available, implementation approaches. (3) Evaluation of LTBI screening programme and measures used to assess outcome per report (e.g., identification of LTBI or active TB, treatment initiation and completion, economic evaluations, acceptability).

Ethical committee review or informed consent were not required for this study as data were collected exclusively from published sources.

## Results

21 countries met the inclusion criteria; Australia, Austria, Belgium, Canada, Denmark, Finland, France, Germany, Hungary, Ireland, Israel, Italy, Japan, The Netherlands, New Zealand, Norway, Spain, Sweden, Switzerland, United Kingdom of Great Britain and Northern Ireland and the United States of America. Overall, we identified 2118 studies. After deduplication, 1782 unique studies were identified, 194 studies underwent full text screening, and 41 studies met inclusion criteria.

### Article characteristics

Characteristics of the 41 included studies are summarised in Table 1. Of the 41 studies, 12 reported data from England; eight from Italy; five from Canada; four from Norway; two each from Australia, Germany and Sweden; and one from Switzerland. There were two European Centre for Disease Prevention and Control (ECDC) review studies that summarised data from the Netherlands, Czechia, Portugal and Spain. Of the 41 studies, 24 highlighted barriers or success factors, 31 investigated epidemiological data, and 11 completed an economic evaluation of the LTBI programme. Eight studies evaluated programmes in a primary care setting (seven of these from England), seven did not specify a setting, six were from reception centres, five in new arrival or specialist migrant clinics, four were schools, three were interview-based studies, three based in TB/infectious disease centres, two in public health services and one each from a nurse-led clinic, homeless migrant clinic and one comparison study of different community settings. Twelve included studies were published in 2019, six in 2018, four in 2020 and 2022, three in 2021, two each in 2010 and 2017, and one each in 2005, 2006, 2009, 2011, 2013, 2014, 2016 and 2023.

### Epidemiological analyses

Table 2 gives summary of the larger studies by the countries to meet the inclusion criteria, with all identified studies available in S2 Table.

### Primary care-based programmes

Eight studies discussed primary care-based programmes, with seven of these being English studies. England implemented an national LTBI screening programme in 2015/16 which was initially predominantly based in primary care and has since transitioned to being predominantly based in secondary care TB services [38]. Between 2015–22 there was a decreasing positivity rate from 22% to 16% of IGRAs. Of the 45% of people for whom treatment completion data was available, 75% completed treatment [38]. In England, primary care 'Flag 4' registrations were used to identify people potentially eligible for LTBI screening. Flag 4 identifies those who have registered with a previous address outside the UK or who have spent more than 3 months outside of the UK. A 2014 study found that 48% of Flag 4 registrations were people from countries of TB burden >150/100,000 and that people from high incidence TB burden countries took 619 days to register with a GP, compared to a median of 181 days [31]. A 2018 study found a large variation (0–88%) in screening uptake between GP surgeries and that 103 people out of 719 with positive IGRAs did not attend for follow up after receiving their positive test result [28].

### Reception centres

Six studies published findings from TB screening services based in reception centres. A German study, focusing on asylum-seeking children in reception centres, diagnosed 58/968 (6.0%) children with LTBI and 8/968 (0.8%) with TB disease [41]. Seven of the eight children with TB disease were asymptomatic at the time of diagnosis. This study also found that travel via the Balkan migration route appeared to be a risk factor for TB diagnosis [41]. In Norway, all asylum seekers were identified as eligible for latent TB screening, which was performed via the national reception centre [50,63]. Two Norwegian studies found low levels of specialist review and treatment following positive TSTs, with organisational

**Table 1. Summary of basic study characteristics.**

| Author (Year) | Country of data reporting | International/national/local | Setting | Main theme of extracted data*: | Population summary | Data collection period |
|---|---|---|---|---|---|---|
| Sawka et al. 2019 [21] | Australia | Local | Schools | EP, EC | Newly arrived school children from 'high risk' countries of origin | 2013 - 2017 |
| Hall et al. 2020 [22] | Australia | Local | Members of Indian and Pakistani communities | B/E | People born overseas, identifying as part of the Indian or Pakistani communities | Not specified |
| Pépin et al. 2022 [23] | Canada | Local | New arrivals clinic | EC | Adults and child asylum seekers or refugees with status obtained prior to arrival | 1997 - 2020 |
| Brassard et al. 2006 [24] | Canada | Local (selected schools) | Schools | EP, EC | Newly arrived children aged 4–18 years | 1998 - 2003 |
| Minodier et al. 2010 [25] | Canada | Local | Schools | EP, B/E | Immigrated school children | 1997 - 2007 |
| Rennert-May et al. 2016 [26] | Canada | Local | New arrivals clinic | EP, B/E | Newly entered government sponsored refugees of all ages, from all countries of origin | 2009 - 2011 |
| Milinkovic et al. 2019 [27] | Canada | Local (Ontario) | Interviews with service planners, providers and clients | B/E | People who have migrated to Canada | 2017 − 2017 |
| Loutet et al. 2018 [28] | England | Local | Primary care | EP | Migrant populations born or lived in countries of high incidence (>150/100,000) or SSA and entered the UK in the last 5 years. | 2014 - 2015 |
| Zenner et al. 2017 [29] | England | Local | Not specified | EP | New migrants <35 years through registration with GP or port of entry referral arriving from 'high incidence' countries of origin. | 1989 - 2001 & 2009 −2012 |
| Usemann et al. 2019 [30] | Switzerland | Local | Kindergartens and schools | EC | Children born abroad or migrated in the last 12 months from select countries. | 2001 - 2015 |
| Panchal et al. 2014 [31] | England | Local | Primary care | EP | All new migrants of all ages registering with a GP from a previous abroad address. | 2000 - 2010 |
| Berrocal-Almanza et al. 2022 [32] | England | National (selected sites) | Primary care | EP | Migrants aged 15–35 years arrived in the UK within 5 years from a country of TB incidence >150/100,000 population. | 2011 - 2018 |
| Berrocal-Almanza et al. 2019 [33] | England | National (England, Wales and NI) | Primary care | EP | Migrants aged 16–35 years arrived in the UK within 5 years from a country of TB incidence >150/100,000 population, identified through GP registration. | 2011 - 2014 |
| Berrocal-Almanza et al. 2019 [34] | England | Local (London) | Interviews with community-based organisations and public sector stakeholders | BE | Migrants aged 15–35 years, arrived in the UK within 5 years from a country of high TB incidence or >6 months living | Not specified |
| Gasmelseed et al. 2022 [35] | England | Local (Croydon) | Nurse-led screening clinic | EP, B/E | Migrants aged 16–35 years arrived in the UK within 5 years from a country of TB incidence >150/100,000 population, identied throug GP registration. | 2019 - 2022 |
| Williams et al. 2020 [36] | England | Local | Paediatric infectious diseases clinic | EP | Unaccompanied asylum-seeking children, newly entered and referred for screening | 2016 - 2018 |
| Ikram et al. 2019 [37] | England | Local (London) | Primary care | BE | Recent migrants to the UK | 2017 - 2018 |
| Public Health England 2021 [38] | England | National | Primary care | EP | Migrants aged 16–35 years arrived in the UK within 5 years from a country of birth or lived >6 months in a country of TB incidence >150/100,000 population. | 2015 - 2020 |

*(Continued)*

| Author (Year) | Country of data reporting | International/ national/ local | Setting | Main theme of extracted data*: | Population summary | Data collection period |
|---|---|---|---|---|---|---|
| Pareek et al. 2013 [39] | England | Local (London) | Primary care | EP, EC | Foreign-born migrants aged ≥16 years, entered within 5 years from a country of incidence ≥40/100,000 population or all countries + TB symptoms. Identified through GP registration | 2008 - 2010 |
| Pareek et al. 2011 [40] | England | National (Three centres) | TB centres | EP, EC | Foreign-born migrants aged ≤35 years, entered within 5 years from countries of all incidences. | 2008 - 2010 |
| Mueller-Hermelink et al. 2018 [41] | Germany | National (selected sites) | Reception centres | EP | All asylum-seeking children aged 3–15 years. | 2015 - 2016 |
| Thee et al. 2019 [42] | Germany | Local | Unclear | EP | Unaccompanied minor refuges, <18 years through local authorities and charity organisations. | Not specified |
| Russo et al. 2023 [43] | Italy | Local | Infectious diseases hub hospital | EC | Asylum seekers and undocumented migrants, arrived within 5 years, countries of all incidence | 2019 - 2022 |
| Barcellini et al. 2018 [44] | Italy | Local | Centralised TB screening centre for homeless migrants | EP | All foreign-born migrants arriving at reception centres ages <36 years | 2009 - 2017 |
| Carvalho et al. 2005[1] | Italy | Local (Brescia) | Specialist immigrant clinic | EP, B/E | Undocumented immigrants from countries of incidence >50/100,000, entered recently with the intention to live or work in the region for at least six months, aged 18–35 years. | 2001 − 2001 |
| Villa et al. 2019 [45] | Italy | Local (Milan) | Reception centres | EP | Asylum seekers residing in Milan's reception centres | 2016 - 2017 |
| Bonvicini et al. 2018 [46] | Italy | Local | Specialist immigrant clinic | EP, B/E | Irregular immigrants (those not entitled to a GP), ≥ 15 years | 2012 - 2013 |
| Bordin et al. 2022 [47] | Italy | Local | Specialist migrant clinic | EP | All asylum seekers residing in the region and evaluated by the service for migrants. | 2015 - 2017 |
| Marrone et al. 2020 [48] | Italy | Local (Rome) | Reception centres | EP | Unaccompanied immigrant minors (<18 years) on arrival to reception centre | 2013 - 2019 |
| Pontarelli et al. 2019 [49] | Italy | Local (Brescia) | Reception centres | EP | Asylum seekers in accredited reception centres. | 2015 - 2016 |
| Harstad et al. 2010 [50] | Norway | National | Reception centres | EP | All asylum seekers, identified via reception centres. | 2005 - 2006 |
| Haukaas et al. 2017 [51] | Norway | National | Not specified | EC | Entry screening for migrants from countries of ≥40/100,000, <35 years and planning on staying >2 years. | Not specified |
| Winje et al. 2019 [52] | Norway | National | Not specified | EP | Migrants from the top 10 TB source countries according to Norwegian data | 2008 - 2011 |
| Harstad et al. 2009 [50] | Norway | National centre | National Reception Centre | EP | All asylum seekers >15 years, identified via reception centres. | 2005 - 2006 |
| Shedrawy et al. 2021 [53] | Sweden | National | Not specified (pre-hospital) | EC | Modelling study using available data for migrants from all countries of incidence | 2015 - 2019 |
| Nederby Öhd et al. 2021 [54] | Sweden | Local | Primary care | EP | Asylum seekers from countries of high incidence (>100/100,000) or resided in high-risk environment (e.g., prisons or refugee camps). | 2015 − 2015 |
| Spruijt et al. 2019 [55] | The Netherlands | National (selected sites) | 5/25 Public health services, 2 urban and 3 rural. | EP, B/E | Immigrants, not applying for asylum, from non-EU countries with a TB incidence of >50/100,000 population, of all ages, with an intended stay of at least six months, | 2016 − 2016 |

*(Continued)*

**Table 1.** (Continued)

| Author (Year) | Country of data reporting | International/national/local | Setting | Main theme of extracted data*: | Population summary | Data collection period |
|---|---|---|---|---|---|---|
| Spruijt et al. 2020 [56] | The Netherlands | National (selected sites) | Six settings evaluate, ran by public health services | EP, B/E | Eritrean migrants entered within 10 years. Moderate risk (89/100,000) but high risk group in The Netherlands. | Not specified |
| Spruijt et al. 2019 [57] | The Netherlands | National (selected sites) | Public health services | EP, B/E | Asylum seekers >12 years from countries >200/100,000 incidence | 2016 - 2017 |
| ECDC, 2018 [58] | Europe | Multi-national | European report using national level data | EP | The Netherlands, Czechia, Portugal and Spain | 2005 - 2014 |
| ECDC, 2018 [59] | Europe | Multi-national | European report using national level data | EC | The Netherlands, Czechia, Portugal and Spain | 2005 - 2014 |

*Theme area abbreviations: epidemiolocal (EP), economic (EC), barriers/enablers (B/E).

Any programmes operating sub-nationally are categorised as local

factors and reluctance of clinicians to treat those without a permanent visa cited as potential explanations [50,63]. Three Italian reception centre screening studies were published, where TB screening services are required for all asymptomatic individuals from high-endemic countries (TB incidence >100/100,000) staying at reception centres for at least six months [45,48,49,62]. Marrone *et al.* reported country of origin to be a strong predictor of LTBI diagnosis, particularly sub-Saharan Africa [48].

### New arrival and specialist migrant clinics

Two Canadian studies described TB screening services delivered in new arrivals clinics [23,26]. Pépin *et al.* describe the findings of a TB screening programme for refugees, roughly half of whom were from Afghanistan [23]. Of the 3544 refugees seen at the clinic, 441 were diagnosed with LTBI (12.4%) and 374/441 (84.8%) completed treatment. There was a 69.3% compliance with the screening and treatment cascade for all refugees settled in the region, including those who did not present to the clinic. Potential reasons for relatively high compliance rates compared with other TB screening programmes included embedding of the programme in an integrated, family based health care setting, close collaboration with community services and easy access to translators [23]. Three Italian studies described TB screening in specialist migrant clinics [46,47,68]. Of 2,486 asylum seekers screened for TB at the Migrant's Service in Verona, 28.8% were diagnosed with LTBI [47]. Of the screening population, 91.1% were male and 81.6% were African. The 30.3% of people who started treatment but did not complete it, stopped due to medical conditions or transfer to another centre. The study found similar levels of adherence to screening (83.2–94.3% between asylum seekers of different nationalities, but highly variable treatment completion rates, with 42.3% and 49.1% of migrants from Bangladesh and Pakistan respectively completing treatment, compared to 82.8% and 84.9% of migrants from Ghana and Gambia respectively completing treatment [47]. Bonvicini *et al.* who enrolled 368 irregular immigrants (those without a valid resident permit and entitlement to a general practitioner) for TB screening in Northern Italy, commented that had they excluded irregular immigrants from countries of a low TB incidence, they would have excluded nearly half of the study population and missed 80% of TB diagnoses [46].

### Schools

Four studies conducted TB screening programmes in schools, one in Australia screening children from 'high risk' countries of origin, two in Canada screening newly arrived and immigrated school children and a study in Switzerland where children born abroad or moved within the last 12 months from selected countries were screening in a routine school health

**Table 2. By country, a summary of the larger and more recent literature examining latent TB screening programmes or feasibility studies, alongside current screening guidelines/practices for that country. A complete summary of the literature can be seen in S2 Table.**

| England: Primary care-based screening programme | | | | |
|---|---|---|---|---|
| **Population screened** | **Recruitment method** | **Results** | **Notes** | **Current guidance/programme status** |
| Aged 16–35 years arrived in the UK within 5 years from a country of birth or lived > 6 months in a country of TB incidence >150/100,000 population | GP Flag-4 coded registration (indicates last address was abroad) | IGRA used. Public Health England reported 22,221 tests were received in 2020 and the positivity rate decreased through 2015–2022. More women tested than men, but men had higher positivity rates. 56% of those tested were from India and Pakistan. Treatment completion data available for 45% of those with a positive test and completion rates of 75% of those who started treatment. Positivity rate declined from 22% in 2015–2016 to 16% in 2019–2020 [38]. | There are multiple studies published in England comparing pre- and post- screening evaluations. A 2014 study evaluating all Flag 4 registrations found 28438 (48%) registrations were from countries with a TB incidence ≥150/100k. Median time of GP registration was 181 days from GP entry, but 619 days from people arriving from high incidence countries. 29.2% of foreign-born TB could have been preventable with screening. >50% of foreign-born TB was not preventable with GP based screening, due to failure to register with GP. Delay to Flag-4 registration was nearly 3x longer in immigrants progressing to active TB [31]. | Latent TB screening programme in place. Initially most CCGs testing in primary care and treated in secondary care. Now half of CCGs operate this model, with some testing and treating in TB services (secondary are or community care) and others offering a mixed model [38]. Eligible population for screening: New-entrants born or spent more than 6 months in a high TB incidence country (≥150/100,000 or Sub-Saharan Africa) aged 16–35 year, entered within the last 5 years with no previous history of TB or LTBI. |
| **Australia: School-based screening study, post-migration visa application screening programme** | | | | |
| Newly arrived school children from 'high risk' countries of origin, 2013–2017 [21] | Via the school system | TST used. 4736 tests completed. 17.9% of TSTs were positive. 15.9% of the students referred completed a course of treatment.10.7% of those who started treatment did not finish or were transferred. | This was an abstract including an economic evaluation concluding that the programme appeared to be cost effective but reasons for screening refusal and limited treatment adherence need further evaluating. | Recommendation for post-arrival screening in migrants most likely to benefit. Priority to migrants from countries with incidence of ≥100 per 100,000 per population aged ≤35 years or ≥35 years with risk factors for progression. Consider screening incidence 40–99/100,000 as resourcing permits. Conducted for offshore visa applicants staying >12 months as a condition of grant [60]. |
| **Germany: reception centre testing** | | | | |
| All asylum-seeking children aged 3–15 years, 2018 [41]. | Asylum seeker reception centre before moving on to temporary accommodation | Hamburg: TST. Bochum: < 5 years TST, > 5 years initially IGRA due to TST shortage 66/968 (6.8%) of screened children had TB infection (58 LTBI, 8 active TB). LTBI prevalence was similar in children from high (Afghanistan) and low (Syria) incidence countries (8.7% vs 6.4%). | Children under the age of 6 years were at higher risk of progression to active TB (19% vs 2% respectively, p=0,07). 7/8 children with active TB were asymptomatic at the time of diagnosis. The risk of developing TB in adult asylum seekers is higher in comparison to country of origin. The Balkan route for migration appeared to be a risk factor for TB. | There is no national screening programme for migrants. IGRAs are used to pre-screen for clinical TB in some cases. Responsibility for any conducted screening sits within the community services. Strategies, resourcing and procedures vary [61]. |
| **Italy: methods vary by region with most published literature relating to specialist migrant clinics or reception centres** | | | | |
| Homeless migrants, asylum seekers, undocumented migrants, irregular immigrants | Generally via reception centre or social workers | TST or TST with confirmatory IGRA used. The largest study tested 9486 homeless migrants applying to shelters, 2666 had positive TSTs and attended for confirmatory IGRAs, of which 50.2% were positive. 72 people diagnosed with active TB. The odds of being IGRA positive were higher in people from countries with an incidence of 51–150 per 100,000 population vs 151–250 per 100,000 population [44]. | A study testing 368 irregular immigrants (those not entitled to a GP) commented that had they excluded immigrants from countries with a low TB incidence they would have excluded 48% of subjects and missed 80% of TB cases [46]. | The following guidelines are provided by the Italian National Institute for Health, Migration and Poverty (NIHMP): TST or IGRA (the latter particularly in previously vaccinated individuals) is recommended for all asymptomatic individuals from high-endemic countries (TB incidence >100/100,000) staying at reception centres for at least six months [62]. |

*(Continued)*

| England: Primary care-based screening programme | | | | |
|---|---|---|---|---|
| **Population screened** | **Recruitment method** | **Results** | **Notes** | **Current guidance/programme status** |
| Norway: National reception centre | | | | |
| All asylum seekers >15 years, identified via reception centres. 2009 [63]. | On arrival all asylum seekers are referred to the national reception centre (NRC) for management of immediate medical needs and compulsory TB screening. | TST used. Of 5112 asylum seekers 91% were eligible for inclusion. 97.5% were tested. 46% had a positive TST. Only 16% of those with a strongly positive TST were reviewed by a specialist. | Informal reasons for low follow-up after positive test results were given as short stay at the NRC, high workload at the central TB clinic and uncertainty around length of stay of asylum seekers in country. A 2019 study evaluating the number needed to treat (NNT) and number needed to screen (NNS) to prevent one case of TB found a closer correlation between NNT and Norwegian notification rate than NNT and WHO incidence rate in country of origin [52]. | All asylum seekers and refugees to Norway, regardless of their country of origin, are required to undergo a TB test within two weeks of their arrival in the country. These tests normally happen at an initial health assessment. There is follow-up voluntary health check after 3 months, Other migrants who intend to stay for over three months in Norway are tested for clinical TB if they come from a country with a TB incidence >40/100,000 population and for latent TB with an IGRA if they come from a country with a TB incidence > 200 per 100,000 population [64]. |
| Sweden: primary care | | | | |
| Asylum seekers from countries of incidence (>100/100,000) or resided in high risk environment (e.g., prisons or refugee camps). 2021 [54]. | Individuals attended a free voluntary health examination, centralised to selected primary health centres. | IGRA ≥2 years old, and TST or IGRA <2 years 1364/5470 (24.9%) of IGRAs were positive. 358 started treatment with 91% completing treatment. 1371 IGRAs were performed on individuals not eligible based on country of origin but with an additional risk factor. | Prevalence of a positive IGRA was similar between asylum seekers from country-of-origin with a TB incidence of 50–99 per 100,000 population + a risk factor and those from a country with incidence 100–199 per 100,000 population. A 26% treatment inhiation rate was explained by policy recommended treatment in those aged >20 years if a risk factor is present. The study questions the appropriateness of testing people not eligible for latent TB treatment. | Policy recommendation for screening of new arrivals from countries with TB incidence of >100/100,000 population, or those with an incidence of <100/100,000 population and have been in an environment with an increased risk to TB, e.g., prisons or refugees camps, or had close contact with a person with TB [65]. |
| The Netherlands: Public health services | | | | |
| Asylum seekers >12 years old from countries >200/100,000 incidence, 2019 [57]. | Offered voluntary screening to asylum seekers >12 years living in asylum seeker centres (ASC) from countries with a TB incidence >200 per 100,000 population | IGRA used. 209/719 IGRAs were positive. Of the 209, 178 people were diagnosed with LTBI, 20 were lost to follow up, 4 were diagnosed with active TB and 3 had been previously treated for TB. 129/148 people starting TB treatment finished it. The coverage of LTBI screenings organised at the ASC (average 63%; minimum 50%, maximum 87%) was slightly higher and fluctuated less than the coverage of LTBI screenings organised at the PHS (average 59%; minimum 8%, maximum 96%). | Eritrean clients were interviewed for the qualitative portion of this study. Facilitators were noted as planning screening to coincide with mandatory ASC reporting improved presence and meant that ASC staff members could locate clients in the case of a no-show. Allowing clients to invite friends and family to screening improved attendance. Simple messages from clinic nurses and weekly boxes may have aided treatment completion. | The 2016–2020 plan laid out the objective to supplement or replace radiographic screening (use of chest X-rays to check for clinical TB) with latent TB screen for children and immigrants from high-risk countries (>200/100,000), with regions to develop integrated package of infectious diseases control and health promotion interventions for immigrants and asylum-seekers [66]. |

*(Continued)*

**Table 2.** (Continued)

| England: Primary care-based screening programme | | | | |
|---|---|---|---|---|
| Population screened | Recruitment method | Results | Notes | Current guidance/programme status |
| Canada: New arrivals clinic | | | | |
| Adult & child asylum seekers or refugees (status obtained prior to landing). 2022 [23] | Clients presenting to Sherbrook refugee clinic | TST predominantly used. NNS: 95.1. NNT: 11.9. 8.6% diagnosed with latent TB. 85% completed treatment. Screening was delivered as part of an integrated migrant health package. | This study commented that key factors for success of this program, leading to relatively high compliance and cost-effectiveness were: "*close collaboration with community organizations, integration within a comprehensive package of medical care for the whole family, timely delivery following arrival, shorter treatment through preferential use of rifampicin, and risk-based selection of patients to be treated*" | Post-landing surveillance recommendations made by the Canadian Thoracic Society for screening are based on a thresh-old of 1% risk of developing TB within 5 years. Recommendation (all conditional recommendations except where noted). • All foreign-born people of all ages with conditions associated with a very high risk of TB reactivation (strong recommendation) • Foreign-born people from countries with a TB incidence of ≥ 50/ 100,000 with conditions associated with a high risk of reactivation • Refugees from countries of incidence >50/100,000 aged ≤65 years, as soon as possible and up to two years following arrival. Consider >65 years based on individual. • Foreign-born people from countries of incidence >200/100,000 population with low to moderate risk of reactivation, as soon as possible and five years after arrival. Can consider >65 years based on individual. Recommend against routine TB screening for those born outside of Canda from countries with incidence <50/100,000 [67]. |

appointment [21,24,25,30]. In Australia, 846/4736 school children tested using TST had positive results and two were diagnosed with active TB [21]. Brassard *et al.* reported on a TB screening programme in Montreal where children were tested and the programme cascaded to include families and households of those children diagnosed with LTBI [24]. This study found TST positivity in 542/2524 (21.5%) of children tested and in 211/555 (38.0%) of associates tested, demonstrating the effectiveness of a cascading programme [24]. TST positivity was similar in the second Canadian study, at 777/3401 (22.8) tested school children [25]. In the Swiss study, TB screening was integrated into routine school heath appointments for migrant children from Afghanistan, the African continent, Portugal, Greece, Albania, Western Balkan, Turkey, South and Central America, Russian Commonwealth, and all Asian/Pacific countries, except Australia and New Zealand and 21/1120 (1.9%) of tested children had positive TSTs [30]. Of the 21 children, 14/21 (66.7%) were from the African continent [30].

## TB or infectious disease centres

Three studies reported findings from TB screening programmes conducted in dedicated TB or infectious disease centres [36,40,43]. Pareek *et al.* (2011) evaluated 1229 screened foreign born new entrants to the UK, aged ≤35 years referred to three regional centres in the UK and found a 20% IGRA positivity rate and that 92% of LTBI was detected at screening at an incidence level of ≥150/100,000 including India [40]. Williams *et al.* described a screening programme for unaccompanied asylum-seeking children newly entered to the UK and seen in two paediatric infectious diseases clinics, who were also screened for other infectious diseases [36]. Most attendees were male (88%). TB tests were positive in 55/238 (23.1%) and 3/238 (1.3%) were diagnosed with active TB. There was a high rate of co-infections found, at 10/201 (5.0%) diagnosed with hepatitis B and 27/164 (16.5) positive tests for schistosomiasis [36]. Russo *et al.* reported on an Italian programme screening asylum seeker and undocumented migrants in the Brescia area of Italy using either TST only or two step TST followed by a confirmatory IGRA [43]. In total, 170/595 (28.6%) of migrants were diagnosed with LTBI, with the study finding improved screening completion in the IGRA only sub-group but comparable levels of treatment initiation between IGRA only and the two step screening [43].

## Other settings

In the Netherlands, TB screening for migrants has been conducted by Public Health Services [55–57]. Spruijt et al. conducted a study across five centres for immigrants not applying for asylum from non-EU countries with a TB incidence of >50/100,000 population intending to stay for more than six months [55]. Of 566 people who received tests, 101 (17.8%) were diagnosed with LTBI and three (0.5%) with active TB. Treatment initiation levels were variable between the five centres (29–86%), impacted by the practices of clinicians, where the highest proportion was achieved by the centre which tested only those they would intend to provide treatment to. Potential reasons for low treatment initiation were identified as clinician concerns, including the public health benefit of treating people who may return to high incidence countries or move on from The Netherlands [55]. In Milan, Italy, a screening programme was provided for all foreign-born individuals registering for accommodation services with no restrictions placed according to TB incidence in country of origin [44]. Of 11,585 people who applied for accommodation and were screened for TB in Milan between 2009 and 2017, 9,486 (81.2%) were migrants. 2666/9486 (28.1%) had positive TSTs and received a confirmatory test using IGRA, 1339 (50.2%) of which were positive. The study presented the top ten countries of origins of migrants according to IGRA positivity rates, and three of the countries with the highest rates had TB incidence of ≤100/100,000; Eritrea, Morocco and Romania. The authors suggested that screening based on country of origin incidence is overly restrictive and vulnerable groups or country-specific factors should be considered in screening and treatment guidance [44].

## Economic analyses

The findings of all ten studies performing cost analyses are summarised in Table 3. Evaluations ranged from cost and estimated saving calculations to in-depth modelling studies. All studies concluded that latent TB screening for migrants was cost-effective within certain eligibility criteria specific to each study.

Pareek et al. built a decision analysis model based on screening immigrants 35 years and younger in three centres in the UK using IGRA [40]. The study found the most cost-effective screening options were adults aged 16–35 years from a country of origin with a TB burden of >250/100,000 population and >150/100,000 population with incremental cost effectiveness ratios (ICERs) of £17,956 and £20,819 respectively. However, the former cutoff for screening would identify 29% of LTBI in the migrant population whilst the latter would identify 92%, therefore recommending the lower cutoff. The same study found screening in <16 years at a threshold of TB burden <40/100,000 population to be cost effective and made the recommendation to screen all immigrant children regardless of country of origin incidence due to small population numbers and importance of prioritising children for tuberculosis control [40].

**Table 3. Economic evaluation of latent TB screening programmes for migrants in high-income, low TB incidence countries with high levels of migration.**

| Country | Author, Year (ref) | Setting and population | Test used | Summary |
|---|---|---|---|---|
| Australia | Sawka et al. 2019 [21] | School based programme: Newly arrived school children from 'high risk' countries of origin. | TST | This conference abstract found **latent TB screening to be cost-effective** in an evaluation of a school-based screening programme, screening 4736 newly arrived children from 'high'-risk countries in multiple South Australian schools using TSTs. Screening facilitated an estimated saving of $155,736 AUD whilst the programme cost $102,276 AUD. |
| Canada | Pépin et al. 2022 [23] | New arrivals clinic: Adults and child asylum seekers or refugees with status obtained prior to landing. | TST, with ad hoc IGRAs | This Canadian study found latent TB screening to be **beneficial, effective and cost-effective** with a benefit cost ratio of 2.03. The study screened 5131 child and adult refugees in a regional LTBI screening programme as part of an integrated migrant health package. Screening cost per person was $95 with LTBI treatment costing $590 pp. There was an overall treatment cost $16056 and each case of TB averted represented a saving of $32631 over 30 years. Amongst nationals of the 20 countries where refugees came from, incidence of TB decreased from 68.2 (1997–2008) to 26.3 per 100,000 person-years (2009–2020). When IGRA was used more frequently later in the study, NNS increased but NNT decreased. |
| Canada | Brassard et al. 2006 [24] | School based: children aged 4–18 | TST | This study screened children and their associated adults. 3710 people were offered tests. Screening was found to be **cost effective** with a net saving of $363,923 ($72,785 per year). Associate investigation alone contributed $95530 of savings ($19,106 per year). The cost of the programme was $193,461. |
| Switzerland | Usemann et al. 2019 [30] | Kindergartens and schools: children born abroad or moved in last 12 months from select countries. Integrated into routine school health appointment. | TST | This study found that TB screening would be **cost-effective** for population groups with a latent TB prevalence of >14% at a progression rate of 5%. Groups with lower prevalence are cost effective if progression rates to active TB are higher. |
| England | Pareek et al. 2013 [39] | Port-of-arrival: Migrants aged 16–35 years arrived in the UK within 5 years from a country of birth or lived >6 months in a country of TB incidence >150/100,000 population. | Comparison | This is was a port-of-arrival study, evaluating stratified TST, unstratified TST, QFN-FIT and T-SPOT. There was an association between both age TB incidence in country of origin with screening positivity. The analysis found the programme would avert between 15.6 and 28.8 cases of active TB over 20 years with costs of £594,956.9 and £1,530,303.0 for TST or IGRA, or IGRA alone, respectively. The study found **single-step IGRA could cost-effectively eliminate mandatory chest X-ray on arrival.** |
| England | Pareek et al. 2011 [40] | TB services: Foreign-born migrants aged ≤35 years, entered within 5 years from countries of all incidences. Referred to TB services through port-of-entry screening systems, health-protection units, or after registration with primary-care services. | IGRA | This study found that the two most cost-effective strategies were to screen individuals from countries with a tuberculosis incidence of more than 250 & 150/100,000 population. The incremental cost-effectiveness ratios were £17 956 and £20 819 per prevented case of tuberculosis. Screening at ≥150/100,000 identified 92% of infected immigrants, as opposed to 29% of infected immigrants at the 250/100,000 threshold. Screening at >40/100,000 found 100% of latent infections at an ICER of £29,403. Screening all migrants had an ICER of £101,938. **The study recommended the most cost-effective strategy involved screening <16 years at ≥40 and >250/100,000 16–35 year, however given the significantly higher yield at >150/100,000 for only a small additional cost, this approach should be used.** |
| Italy | Russo et al. 2023 [43] | Recently arrived asylum seekers and undocumented migrants, arrived within 5 years, countries of all incidence. referred by social workers and educators. | TST & IGRA vs IGRA | This was an Italian study evaluating the two step TST then IGRA vs IGRA cost effectiveness. **Both were found to be cost-effective**, with the two-step method being more cost-effective. There was a lower initial cost with TST then IGRA vs IGRA alone, at €57.62 pp vs €74.84 pp. IGRA was noted to have an operational advantage with patients assigned to IGRA alone more likely to complete the screening cascade, which the study concludes may justify the additional cost. The study did not consider social or indirect costs and there was no difference found in treatment initiation rates between the two groups. |

*(Continued)*

| Country | Author, Year (ref) | Setting and population | Test used | Summary |
|---------|---------|---------|---------|---------|
| Norway | Haukaas et al. 2017 [51] | Entry screening for migrants from countries of >40/100,000, <35 years and planning on staying >2 years. | Multiple | This was a Norwegian study using cohort simulation, Markov modelling and a combined decision tree to evaluate four screening and treatment scenarios: (1) "No LTBI screening," (2) "TST+IGRA" (screening with TST first, then IGRA if TST is positive), (3) "IGRA" (no TST), and (4) "IGRA risk," (IGRA only for those with known risk factors for reactivation). **The study found screening options IGRA and IGRA risk to be cost effective. Screening option TST+IGRA was not cost effective.** The study findings suggested that the greatest costs come from screening and treating TB disease, not from screening for and treating LTBI. The increased sensitivity of IGRA substantially reduced the amount paid per avoided case of TB. |
| Sweden | Shedrawy et al. 2021 [53] | Not specified (pre-hospital) | IGRA | Markov modelling to assess the cost effectiveness of the current LTBI screening programme compared to no screening used ICER in terms of societal cost per QALY. **Screening aged 13–19 years was the most cost effective, ages 0–12 and 20–34 were moderately cost effective and aged above 34 years not cost effective (mainly due to most of this age group being ineligible for treatment).** Cost effectiveness could be improved by targeting migrants from high incidence countries and/or increasing treatment initiation rate. |
| The Netherlands, Czechia, Portugal and Spain | European Centre for Disease Prevention and Control, 2018 [53] | Four low TB burden countries in the EU [59] | Multiple | This study evaluated cost effectiveness of LTBI screening using a deterministic model for a variety of high-risk groups and for TST vs IRGA vs two-step TST/IGRA. **Two-step testing was most cost-effective from a health care perspective and either only IGRA or TST/IGRA was most cost-effective from the societal perspective due to the single visit required**. Cost-effectiveness in migrants increased in groups of higher of country-of-origin TB incidence. Migrants included all first-generation migrants (including refugees and asylum seekers) from areas with TB incidence of >50/100,000 population. |

Both Haukaas et al. (Norway) and Shedrawy et al. (Sweden) used Markov modelling [51,53]. The Norwegian study compared four screening scenarios for migrants from countries of TB burden ≥40/100,000, <35 years and planning on staying in Norway >2 years [51]. The study found that screening migrants using IGRA was cost effective and highlighted that the biggest cost in a TB programme comes from screening and treating TB disease, not LTBI [51]. The 2021 Swedish study calculated ICERs in terms of societal cost per quality-adjusted life year and found that screening aged 13–19 years was most cost effective, whilst screening above 34 years was not, mainly due to treatment ineligibility [53]. An ECDC report evaluated cost effectiveness of LTBI screening for migrants of countries of TB burden >50/100,000 population in four European countries and found that two step TST was most cost effective from a healthcare perspective, whilst performing a single test was most cost effective from a societal perspective [59]. Russo et al., Italy, noted that whilst there was a cheaper initial cost with TST, the single visit required with the IGRA had an operational advantage and people tested using this method were more likely to complete the screening cascade [43].

Studies in Australia and Canada conducted in 2019 and 2006 respectively found school-based screening programmes to be cost effective through facilitating estimated savings over the cost of the programmes [21,24]. The Canadian study by Brassard et al. screened associated adults of children with positive TSTs which facilitated 26.2% of the net cost savings [24].

## Barriers and enablers

The most cited success factors were related to structural cohesiveness: strong co-ordination and integration of health care services (including 'one-stop-shop' approaches), collaboration with community partnerships, co-ordination of care with social workers or accommodation staff, and cohesive and streamlined services. Other success factors included delivering services in community settings, taking a whole family-and-friends approach to screening, access to interpreters, ensuring appointment convenience and providing information (including through methods such as leaflets or counselling).

The commonly cited barriers to successful programme delivery were lack of understanding of latent TB and misconceptions, service fragmentation and resource capacity. Service barriers included lack of communication between organisations, data-gaps or sub-optimal data sharing between organisations and treatment or screening delays. Divergence of practice through lack of knowledge of frontline staff, lack of clear treatment guidelines following screening and clinician hesitancy to treat were also cited. Other pertinent barriers include low/no/delayed registration with primary care, temporary nature of accommodation. S3 Table details the frequency with which specific facilitators and barriers were mentioned by the included studies.

## Discussion

Achieving the WHO end TB strategy target of reducing TB incidence by 95% by 2035 requires countries with a low TB burden to adopt effective approaches for both active and latent TB infection [1]. Our rapid review synthesised evidence on the effectiveness, cost effectiveness and implementation challenges of latent TB screening programmes for migrants in high-income, low burden countries. The evidence consistently shows that such programmes can be implemented effectively and deliver long term health and economic benefits, though persistent barriers to implementing programs in these population groups remain.

Successful LTBI screening programme were characterised by strong integration of healthcare, community and social services. These factors were important in improving screening uptake and continued participation in the treatment cascade. In contrast, service fragmentation, poor communication between organisations and data sharing limitations were commonly cited barriers. With some migrant populations, such as asylum seekers and refugees, being highly transient, it is vital that there are not missed opportunities due to poor collaboration between services. Stigma among migrants and misconceptions around TB also hindered engagement. Designing services that incorporate community partnerships, offer interpretation support and provide multiple entry points for screening (e.g., primary care, reception centres, community clinics), appear central to improving programme reach and adherence, particularly in the most vulnerable migrant populations.

Economic evaluations consistently demonstrated long-term cost savings across diverse settings, including schools, integrated specialist migrant clinics, local and national programmes. Cost-effectiveness varied according to eligibility and design, emphasising the importance of implementing new programmes with context-specific features and a good baseline knowledge of the populations likely to be targeted and the specific challenges faced. While targeting migrants from the highest TB burden counties was the most cost effective strategy, this approach missed a significant proportion of people with latent TB. Several studies showed that expanding eligibility to include lower-burden countries can remain cost effective while enhancing overall impact.

In recent years, increases in TB mortality have been seen in both high and low burden countries, and 2018 United Nations (UN) reduction targets were not achieved [69,70]. England, Wales, Scotland, The United States of America and Canada have all seen rises in TB incidence [12–14,71,72]. Our findings align with international guidance from the UN which emphasise the importance of national TB strategies with multi-sectoral and integrated, community-based approaches [73]. International guidance regarding latent TB screening is variably implemented. Where guidance is in place, actual programme existence and design are even more variable.

Low burden countries must continue to strength domestic efforts while contributing to global TB elimination through collaborative effort, international alignment and outcome sharing.

Incorporation of recent advancements in LTBI screening and management have the potential to enhance screening programme effectiveness. Shorter treatment regimens, such as three months of rifampicin and isoniazid (3HR) improve adherence compared to traditional six or nine month regimens [74,75]. New *Mycobacterium tuberculosis* antigen-based skin tests (TBST) which appear to provide comparable specificity and sensitivity to IGRA testing, while retaining the operational simplicity of TST, though current evidence is limited [76]. The implementation of TBST in

settings currently using TST or TST followed by confirmatory IGRAs has the potential to reduce false positives and increase cost effectiveness [76].

This study makes several key recommendations for low TB burden, high income countries with high net migration. Investment in latent TB screening programmes by high-income countries with a low TB burden should be prioritised as a core component of TB elimination strategies. Programmes should use flexible, multi-setting screening models to minimise missed opportunities.

There should be strong integration between health services and community organisations to mitigate against the intersecting social and structural barriers faced by migrants. Programmes should be tailored to the specific needs of their vulnerable populations to support diagnosis, treatment and integrated care of these population sub-groups. Shorter treatment regimens and emerging diagnostic tools should be incorporated to improve detection efficiency and treatment adherence.

This study is strengthened by the inclusion of quantitative and qualitative evidence from a range of local, national and international studies globally. However, a limitation of this review is the predominance of studies emerging from select countries; twelve publishing data from England, eight from Italy and five from Canada. 21 countries met our eligibility criteria, but data collated is from a total of nine countries, plus two European reviews. Whilst in some cases, the lack of data is due to no screening programme being present, countries including France, Belgium and Spain have guidance in place regarding TB screening for migrants but no data for inclusion in this review was found. A rapid review methodology may have missed some studies which would have been included a systematic review. The evidence evaluated in this study is regarding high-income, low TB burden, high net migration countries. However, circumstances of individual countries will vary, including the demographics and movement pattern of migrants. When developing a TB screening programme, these individual circumstances should be considered. Widely recognised TB incidence thresholds (>150 per 100,000) for prioritising migrant screening programmes were used in this study. With the need for accelerated efforts to achieve TB elimination, the use of reduced incidence thresholds should be considered, such as the > 100 per 100,000 population for cut off for high incidence laid out in the 2023 TB elimination framework proposed by Migliori *et al.* [77]. Whilst this study is focused on strategies for LTBI detection in low-burden, high income countries, LTBI screening and management is an important component of TB programmes in countries of moderate and high TB incidence.

Latent TB screening programmes for migrants in high-income, low TB burden countries are critical in striving for TB elimination, and reversing the recent increases in TB incidence seen in some nations. Their success depends on cross-sectoral coordination, community integration, and flexible models tailored to vulnerable populations. Building and strengthening these programmes represent an important step in improving the unequitable health outcomes for migrants and striving for TB elimination.

## Supporting information

**S1 Appendix. Search strategy.**
(DOCX)

**S2 Table. Summary of literature evaluating epidemiology of migrants eligible for or screened in a latent TB screening programmes.**
(DOCX)

**S3 Table. Summary of identified barriers and facilitators.**
(DOCX)

## Author contributions

**Conceptualization:** Alice E. Taylor, Hazel Henderson, Eisin McDonald, Peter MacPherson.

**Data curation:** Alice E. Taylor.

**Formal analysis:** Alice E. Taylor, Peter MacPherson.

**Funding acquisition:** Peter MacPherson.

**Investigation:** Alice E. Taylor, Peter MacPherson.

**Methodology:** Alice E. Taylor, Hazel Henderson, Peter MacPherson.

**Project administration:** Alice E. Taylor.

**Supervision:** Hazel Henderson, Peter MacPherson.

**Visualization:** Alice E. Taylor.

**Writing – original draft:** Alice E. Taylor.

**Writing – review & editing:** Alice E. Taylor, Hazel Henderson, Eisin McDonald, Peter MacPherson.

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
