## [Decision Letter · Decision Letter 0]

29 May 2025

Dear Dr. Taylor,

Thank you for submitting your manuscript to PLOS ONE. After careful consideration, we feel that it has merit but does not fully meet PLOS ONE’s publication criteria as it currently stands. Therefore, we invite you to submit a revised version of the manuscript that addresses the points raised during the review process.

We look forward to receiving your revised manuscript.

Kind regards,

Lisa Kawatsu, PhD

Academic Editor

PLOS ONE

 [PM is funded by Wellcome (304666/Z/23/Z). For the purpose of open access, the author has applied a CC BY public copyright licence to any Author Accepted Manuscript version arising from this submission]. 

Additional Editor Comments:

This is a well-written manuscript with a timely focus on LTBI screening among migrants. Please kindly address the comments from the reviewers and make the necessary revisions.

Reviewers' comments:

Reviewer's Responses to Questions

**Comments to the Author**

1. Is the manuscript technically sound, and do the data support the conclusions?

Reviewer #1: Yes

Reviewer #2: Yes

2. Has the statistical analysis been performed appropriately and rigorously?

Reviewer #1: N/A

Reviewer #2: N/A

3. Have the authors made all data underlying the findings in their manuscript fully available?

Reviewer #1: Yes

Reviewer #2: Yes

4. Is the manuscript presented in an intelligible fashion and written in standard English?

Reviewer #1: Yes

Reviewer #2: Yes

Reviewer #1: This a well written, detailed review of an important topic.

I have a few comments:

1. The author should discuss if the definition of "TB high endemic countries" with 150/100,000 pop. new TB cases per year is sound or if the threshold should be lower?

2. Perhaps it should be mentioned in the end of the discussion that also TB high endemic countries could benefit from identifying people with LTB and offer these treatmet.

3. The authors include only industrialized countries and I suggest they include Oman in the review. Oman is conducting a screening program for LTB and hes the advantage that participation is mandatory to receive a work visa. See for instance:

A step forward in tuberculosis elimination: implementing migrant latent tuberculosis screening and treatment in Oman.

Alyaquobi FM et al. IJID Reg. 2025 Mar 19;14(Suppl 2):100614. doi: 10.1016/j.ijregi.2025.100614.

Cost-effectiveness of IGRA/QFT-Plus for TB screening of migrants in Oman.

Al Abri S et al. Int J Infect Dis. 2020 Mar;92S:S72-S77. doi: 10.1016/j.ijid.2020.03.010.

Reviewer #2: Screening migrant populations for latent tuberculosis infection is a common and important challenge in low TB incidence countries. Therefore, this work of the authors is very valuable and the information is very much needed for public health officials in their decision.

This is a well-written manuscript. However, authors have some points to make clear.

1. Why did the authors choose the rapid review method instead of the traditional systematic review method? Screening for LTBI is important, but not as urgent as the situation with the COVID-19 pandemic. In the methods section, please describe the reason of selecting rapid review method.

2. It would be useful to have a convenient table showing the expected rates of TB infection positive, LTBI treatment initiation and treatment completion if LTBI screening were performed on the migrant population. In the results section, please make the table if it is possible.

3. Treatment of LTBI has progressed in recent years. 3HR regimens are more common than 6H. Additionally, a new generation of TB infection diagnostic kits is being developed and now we have new tuberculosis antigen-based skin tests. Could the coming of these new technologies affect future LTBI screening strategies for migrant populations?

4. In conclusion, what is the quantitative impact of LTBI screening for migrants on TB elimination or to achieve the goal of End TB Strategy in low TB incidence coutries? In the discussion section, please describe your prospect.

**Do you want your identity to be public for this peer review?** For information about this choice, including consent withdrawal, please see our Privacy Policy

Reviewer #1: No

Reviewer #2: No

---

## [Author Response · Author response to Decision Letter 1]

1 Jul 2025

The response to reviewers has been uploaded as a file, titled 'Response to reviewers'

---

## [Decision Letter · Decision Letter 1]

17 Sep 2025

Dear Dr. Taylor,

Thank you for submitting your manuscript to PLOS ONE. After careful consideration, we feel that it has merit but does not fully meet PLOS ONE’s publication criteria as it currently stands. Therefore, we invite you to submit a revised version of the manuscript that addresses the points raised during the review process.

We look forward to receiving your revised manuscript.

Kind regards,

Lisa Kawatsu, PhD

Academic Editor

PLOS ONE

Journal Requirements:

**Comments to the Author**

Reviewer #1: (No Response)

Reviewer #2: All comments have been addressed

2. Is the manuscript technically sound, and do the data support the conclusions?

Reviewer #1: Yes

Reviewer #2: Yes

3. Has the statistical analysis been performed appropriately and rigorously?

Reviewer #1: N/A

Reviewer #2: N/A

4. Have the authors made all data underlying the findings in their manuscript fully available?

Reviewer #1: Yes

Reviewer #2: Yes

5. Is the manuscript presented in an intelligible fashion and written in standard English?

Reviewer #1: Yes

Reviewer #2: Yes

Reviewer #1: This is an important review of screening programs for LTB in different countries. Each country is listed separately in 3 table, making the programs easy to understand and compare.

I have a few comments:

1. In the methods section please define the Line 95-6: Rapid review methodologies ?

2. "selected to facilitate the provision of timely insights to inform policy". Describe the criteria for selection

3. Line 113-14 A protocol and search strategy were developed and agreed upon by the research team prior to

literature search. Provide info on. the "protocol"

4. I think that the autors should mention the LTB screening program in Oman: Alyaquobi FM et al. A step forward in tuberculosis elimination: implementing migrant latent tuberculosis screening and treatment in Oman..IJID Reg. 2025 Mar 19;14(Suppl 2):100614. doi: 10.1016/j.ijregi.2025.100614.

5. The discussion is very long and often repetitive. 5 pages which can be reduced to 3 pages.

Reviewer #2: (No Response)

**Do you want your identity to be public for this peer review?** For information about this choice, including consent withdrawal, please see our Privacy Policy

Reviewer #1: No

Reviewer #2: **Yes: ** Kazuhiro Uchimura

---

## [Author Response · Author response to Decision Letter 2]

30 Sep 2025

Response to reviewer letter uploaded as an attachment.

---

## [Editor Report · Decision Letter 2]

20 Oct 2025

The impact of latent tuberculosis screening programmes for migrant populations in high income, low burden countries.

PONE-D-25-15733R2

Dear Dr. Taylor,

We’re pleased to inform you that your manuscript has been judged scientifically suitable for publication and will be formally accepted for publication once it meets all outstanding technical requirements.

Kind regards,

Lisa Kawatsu, PhD

Academic Editor

PLOS ONE
---

## [Editor Report · Acceptance letter]

PONE-D-25-15733R2

PLOS ONE

Dear Dr. MacPherson,

I'm pleased to inform you that your manuscript has been deemed suitable for publication in PLOS ONE. Congratulations! Your manuscript is now being handed over to our production team.

Kind regards,

on behalf of

Dr. Lisa Kawatsu

Academic Editor

PLOS ONE